# On-chip generation and dynamic piezo-optomechanical rotation of single photons

Dominik D. Bühler[1,8], Matthias Weiß [2,3,8] ✉, Antonio Crespo-Poveda[4], Emeline D. S. Nysten [2,3], Jonathan J. Finley [5,6], Kai Müller [6,7], Paulo V. Santos [4], Mauricio M. de Lima Jr. [1] ✉ & Hubert J. Krenner [2,3] ✉

Integrated photonic circuits are key components for photonic quantum technologies and for the implementation of chip-based quantum devices. Future applications demand flexible architectures to overcome common limitations of many current devices, for instance the lack of tuneabilty or built-in quantum light sources. Here, we report on a dynamically reconfigurable integrated photonic circuit comprising integrated quantum dots (QDs), a Mach-Zehnder interferometer (MZI) and surface acoustic wave (SAW) transducers directly fabricated on a monolithic semiconductor platform. We demonstrate on-chip single photon generation by the QD and its sub-nanosecond dynamic on-chip control. Two independently applied SAWs piezo-optomechanically rotate the single photon in the MZI or spectrally modulate the QD emission wavelength. In the MZI, SAWs imprint a time-dependent optical phase and modulate the qubit rotation to the output superposition state. This enables dynamic single photon routing with frequencies exceeding one gigahertz. Finally, the combination of the dynamic single photon control and spectral tuning of the QD realizes wavelength multiplexing of the input photon state and demultiplexing it at the output. Our approach is scalable to multi-component integrated quantum photonic circuits and is compatible with hybrid photonic architectures and other key components for instance photonic resonators or on-chip detectors.

Photonic quantum technologies[1–7] have seen rapid progress and hallmark quantum protocols have been implemented using integrated quantum photonic circuits (IQPCs)[8–13]. For most applications, the on-chip generation of single photons is highly desirable to avoid the inevitable coupling losses when using an off-chip source. For this purpose, quantum emitters[14], for instance semiconductor quantum dots (QDs)[15,16] or defect centers[17–22], are excellent candidate systems. Among these systems, QDs made from (In,Ga,Al)As semiconductor compounds offer several advantages[23–27]: they are extremely bright sources of single photons[28,29] and entangled photon pairs[30,31] which can

[1]Materials Science Institute (ICMUV), Universitat de València, PO Box 22085, 46071 Valencia, Spain. [2]Physikalisches Institut, Westfälische Wilhelms-Universität Münster, Wilhelm-Klemm-Straße 10, 48149 Münster, Germany. [3]Lehrstuhl für Experimentalphysik 1, Universität Augsburg, Universitätsstraße 1, 86159 Augsburg, Germany. [4]Paul-Drude-Institut für Festkörperelektronik, Leibniz-Institut im Forschungsverbund Berlin e.V., Hausvogteiplatz 5-7, 10117 Berlin, Germany. [5]Walter Schottky Institut and Physik Department, Am Coulombwall 4, Technische Universität München München, 85748 Garching, Germany. [6]Munich Center for Quantum Science and Technology (MCQST), Schellingstr. 4, 80799 Munich, Germany. [7]Walter Schottky Institut and Department of Electrical Engineering, Am Coulombwall 4, Technische Universität München, 85748 Garching, Germany. [8]These authors contributed equally: Dominik D. Bühler, Matthias Weiß. ✉e-mail: matthias.weiss@uni-muenster.de; mmlimajr@uv.es; krenner@uni-muenster.de

be elegantly included in the photonic structure during epitaxial growth of a heterostructure[32]. These heterostructures are then ready for monolithic fabrication of IQPCs using advanced cleanroom technology[33–37]. In contrast to monolithic approaches, heterogeneous integration of these QDs and other types of quantum emitters on IQPCs made on material platforms with complementary strengths promise superior performance[12]. Such hybrid devices have been reported on silicon (Si)[38,39], silicon nitride (SiN)[40–44], aluminum nitride (AlN)[45] or lithium niobate (LiNbO₃)[46] IQPCs. However, their fabrication is natively connected to a significant increase of the complexity compared to monolithic routes. Furthermore, the on-chip control of light propagation in photonic elements is crucial. To this end, for instance thermo-optic[47–49], electro-optic[50–52] or acousto-optic[53–56] effects or nanomechanical actuation[57,58] have proven to be viable routes. Among these mechanisms, acoustic phonons are an attractive choice because they couple to literally any system[59] enabling strong optomechanical modulation and dynamic reconfiguration of quantum emitters[60–62]. In the form of radio frequency Rayleigh surface acoustic waves (SAWs)[63] or Lamb waves[64], phonons can be routed on-chip[65–67] and interfaced with integrated photonic elements[53–56,68,69], quantum emitters[70–75] or even superconducting quantum devices[76,77]. The electrical generation of SAWs and their ultralow dissipation offers distinct advantages over local tuning schemes. Thermo-optic, electro-optic or Stark-effect tuning require local electrodes to generate heat or electric fields for each element. SAWs, in contrast however can be piezo-electrically generated by applying a rf voltage to an interdigital transducer (IDT) and the propagating SAW beam modulates any IQPC element or QD in its propagation path. The underlying mechanical tuning mechanism does not induce inherent losses, in contrast to the well-known Franz-Keldysh electroabsorption in electro-optic devices[78]. These unique properties together with the ability to synchronize SAWs and optical pulses[79,80] pave the way towards parallelized control in large scale IQPC networks.

In this work, we report on a piezo-optomechanically reconfigurable quantum photonic device with integrated tuneable QDs, schematically shown in Fig. 1. The validation of fully fledged dynamic single photon routing, single qubit logic and single photon wavelength (de) multiplexing is illustrated in Fig. 1b-d. First, we excite a QD, which emits a single photon into an integrated and dynamically tuneable MZI. Second, we employ a SAW with a frequency of $f_{SAW} \approx 525$ MHz to dynamically route the single photons between the outputs by tuning the time-dependent phase gate which creates a superposition state with a visibility of >0.75 at the operation frequency. Third, dynamic spectral modulation of the QD phase-locked to the tuneable phase gate enables freely programmable spectral multiplexing of single photons. Our demonstrated operation bandwidth of >1 GHz for single photon routing exceeds that reported for state-of-the-art monolithic devices employing electro-optic[52] and nanomechanical[58] tuning by more than a factor 100. Also, our achieved spectral tuning range of the integrated QD of ≥ 0.8 nm is competitive with Stark-effect tuning on this platform[81]. The achieved >1 GHz operation nests our system well in the resolved sideband regime enabling on-chip parametric quantum phase modulation of the QD[74,82,83] and the routed single photons[84].

## Results

### Device layout

Our device is based on a waveguide IQPC monolithically fabricated on a GaAs semiconductor platform[24,26,34]. The piezoelectricity of this class of materials enables the direct all-electrical excitation of SAWs using IDTs[85]. The SAW tunes the integrated MZI via a time-modulated photoelastic effect and switches photons between the two outputs. Importantly, the heterostructure contains (In,Ga)As QDs, one of the most mature semiconductor quantum emitter system[25]. These QDs are established as high quality sources of single photons with high indistinguishability[28,29] and their potential has been unambiguously proven by the implementation of fundamental quantum protocols[86–89].

Our device, schematically shown in Fig. 1a, comprises key functionalities required of an IQPC: it contains integrated, tuneable quantum emitters for in-situ wavelength multiplexed single photon generation. A combination of two static and one programmable elementary single qubit gates rotates the qubit and controls its output state which is detected in a projection measurement. The IQPC itself is based on etched GaAs ridge waveguides (WGs) on an (Al,Ga)As cladding layer. During crystal growth of the semiconductor heterostructure, a single layer of (In,Ga)As QDs is embedded in the active region to provide high-quality built-in, anti-bunched quantum

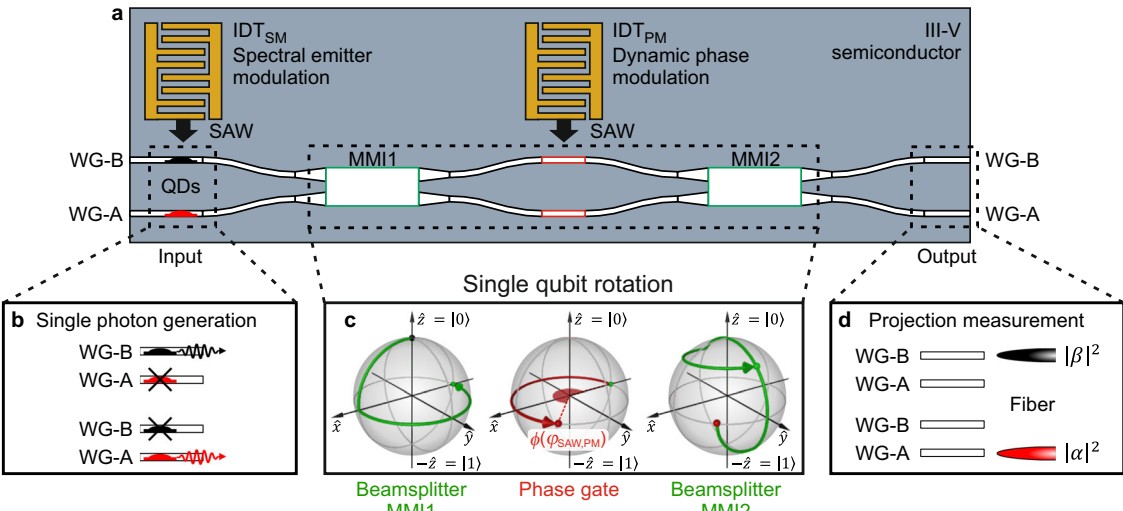

**Fig. 1 | Device and qubit rotation. a** Schematic representation of the dynamically tuneable ridge waveguide integrated quantum photonic circuit (IQPC). The concept, based on a Mach Zehnder Interferometer (MZI) comprises two input waveguides (Input WG-A and Input WG-B) connected to a 2 × 2 multimode interference device (MMI1) whose outputs are connected by two WGs to MMI2 with two output WGs (Output WG-A and Output WG-B). Two interdigital transducers (IDTs) generate SAWs for spectral modulation of single QDs (IDT$_{SM}$) in the Input WGs and optomechanical phase modulation of the MZI (IDT$_{PM}$). **b** Selective single photon generation by a single QD in the input WGs. **c** Single photon rotations shown on Bloch spheres. The evolution of the states is indicated by the green arrows (Beamsplitter gates by MMI1 & MMI2, respectively) and a red arrow (SAW-driven dynamic phase gate). **d** projection measurement of the rotated superposition state by collection and detection from the Output WGs (Output WG-A / Output WG-B) using a lensed optical fiber.

emitters. The photonic circuit consists of two symmetric WGs, referred to as WG-A and WG-B, and two 4-port multimode interference (2 × 2 MMI) beamsplitters. To achieve fully fledged tuneability, our device is equipped with IDTs to generate two SAW beams with the same frequency of $f_{SAW} \approx 525$ MHz.

As explained above, the QDs emit single photons in the input waveguides WG-A and WG-B. Since the QDs are in different WGs, our single photon generation scheme does not strictly correspond to a fully fledged initialization of a photonic qubit. Nevertheless, we apply the corresponding terminology in the following because the state control scheme is equivalent to a qubit rotation. A common representation of a photonic qubit is the so-called rail encoding which can be applied for our IQPC. Here, the qubit states are defined as $|1_A, 0_B\rangle = \begin{pmatrix} 1 \\ 0 \end{pmatrix} \equiv |0\rangle$ and $|0_A, 1_B\rangle = \begin{pmatrix} 0 \\ 1 \end{pmatrix} \equiv |1\rangle$, with indices $A, B$ denoting WG-A and WG-B, respectively[26]. Moreover, we point out that the embedded QDs are dynamically tuneable. They can be strained by a SAW generated by $IDT_{SM}$ and the time-dependent local phase of the SAW $\varphi_{SAW,SM}(t) = 2\pi f_{SAW,SM} t$ programs the emission wavelength of the QD emitted single photons $\lambda(t) = \lambda_0 + \Delta\lambda \cdot \sin 2\pi f_{SAW,SM} t$ used to encode the input[61].

Employing the terminology of rail-encoded photonic qubits, its state vector is controllably rotated on the Bloch sphere in the IQPC as follows. The two input WGs are connected to the first MMI beamsplitter, labeled MMI1, which executes an Hadamard $H = \frac{1}{\sqrt{2}} \begin{pmatrix} 1 & 1 \\ 1 & -1 \end{pmatrix}$ and a Pauli-$Z$, $Z = \begin{pmatrix} 1 & 0 \\ 0 & -1 \end{pmatrix}$, gate operation on the input state. The full beamsplitter gate operation caused by a single MMI can be expressed as

$$MMI = Z \cdot H = \frac{1}{\sqrt{2}} \begin{pmatrix} 1 & 1 \\ -1 & 1 \end{pmatrix} \qquad (1)$$

Thus, MMI1 creates a superposition state from the input photonic qubit, $|0\rangle \to \frac{1}{\sqrt{2}}(|0\rangle - |1\rangle) \equiv |-\rangle$ and $|1\rangle \to \frac{1}{\sqrt{2}}(|0\rangle + |1\rangle) \equiv |+\rangle$, which propagates to the second MMI beamsplitter, MMI2. MMI2 rotates these states $|-\rangle$ and $|+\rangle$ to output states $|1\rangle$ and $|0\rangle$, respectively. Using Eq. (1), the full rotation of the static IQPC can be described by $IQPC_{static} = MMI \cdot MMI = \begin{pmatrix} 0 & 1 \\ -1 & 0 \end{pmatrix} = Z \cdot X$, which corresponds to the combination of a Pauli-$X$ gate (NOT-gate) and a Pauli-$Z$ gate. Since the Pauli-$Z$ gate only affects the phase of the qubit, but not the projection of the qubit on the base states $|0\rangle$ and $|1\rangle$, and the qubit is measured after exiting the IQPC, the $Z$-gate does not influence the outcome of the experiments presented in this work.

The SAW generated by $IDT_{PM}$ optomechanically modulates the optical phase difference between the WGs connecting the two MMIs[54,90]. For this operation, the two arms (length 140 μm) of the MZI connecting the MMIs are separated by 1.5 $\Lambda_{SAW,PM}$, with $\Lambda_{SAW,PM} = 5.6$ μm, being the *acoustic* wavelength of the applied SAW. This geometric separation ensures that the optical phase modulations in the two arms are antiphased, which can be expressed as $\phi_{A,B}(t) = \mp\Delta\phi \sin(\varphi_{SAW,PM}(t))$, with − for WG-A and + for WG-B. Here, $\varphi_{SAW,PM}(t) = 2\pi f_{SAW,PM} t$ is the dynamic phase of the SAW. $\Delta\phi$ is the amplitude of the optical phase modulation, which is given by the strength of the underlying acousto-optic interaction[54]. Most importantly, $\Delta\phi$ can be tuned by the electrical radio frequency (rf) power of the electrical signal $P_{rf,PM}$, applied to the IDT. Thus, the total optical phase shift amounts to

$$\phi(t) = 2 \cdot \Delta\phi \sin(2\pi f_{SAW,PM} t) + \phi_0 \qquad (2)$$

In this expression, $\phi_0$ is a finite optical phase offset introduced by imperfections during nanofabrication. We obtain as the full SAW-driven dynamic phase gate operation

$$R_\phi(t) = \begin{pmatrix} 1 & 0 \\ 0 & e^{i\phi(t)} \end{pmatrix} \qquad (3)$$

After this phase gate operation, MMI2 executes the second beamsplitter gate operation whose output now depends on $R_\phi(t)$.

Most notably, this gate operation allows us to program the output state of the qubit

$$|\Psi\rangle = \alpha(\varphi_{SAW,PM})|0\rangle + \beta(\varphi_{SAW,PM})|1\rangle \qquad (4)$$

$\alpha(\varphi_{SAW,PM})$ and $\beta(\varphi_{SAW,PM})$ are SAW-programmable complex amplitudes obeying $|\alpha|^2 + |\beta|^2 = 1$. Figure 1c exemplifies the respective rotations on the Bloch sphere starting from the initial $|0\rangle$ qubit state propagating through the modulated device. First, MMI1 rotates the input state into the $xy$-plane of the Bloch sphere to the $|-\rangle$ state. Second, the SAW-programmable optical phase shift $\phi(\varphi_{SAW,PM})$, rotates the Bloch vector in the equatorial plane ($xy$-plane) and third, MMI2 rotates the Bloch vector from the $xy$- to the $yz$-plane.

Combining Eq. (1) and Eq. (3), we obtain for the full operation executed by our dynamic IQPC

$$IQPC(\phi) = MMI \cdot R_\phi(t) \cdot MMI = \frac{1}{2} \begin{pmatrix} 1 - e^{i\phi} & 1 + e^{i\phi} \\ -1 - e^{i\phi} & -1 + e^{i\phi} \end{pmatrix} \qquad (5)$$

This full gate operations yield the following states for a single photon created at the input WG-A and WG-B:

$$|1_A, 0_B\rangle \equiv |0\rangle \to \frac{1}{2} \begin{pmatrix} 1 - e^{i\phi} \\ -1 - e^{i\phi} \end{pmatrix} \qquad (6)$$

$$|0_A, 1_B\rangle \equiv |1\rangle \to \frac{1}{2} \begin{pmatrix} 1 + e^{i\phi} \\ -1 + e^{i\phi} \end{pmatrix} \qquad (7)$$

at the Output WG-A and Output WG-B, respectively.

The corresponding probability amplitudes $|\alpha|^2$ and $|\beta|^2$ are obtained via a projection measurement by collecting the output signals of the two waveguides with a lensed fiber.

Note, that truly arbitrary output states can be created simply by adding a second SAW-modulator after MMI2 which can be actively phase-locked to the first SAW-modulator. This straightforward extension creates a dynamic and fully programmable single qubit rotator, an elemental building block of photonic quantum processors[2,91].

Devices were fabricated monolithically on a semiconductor heterostructure grown by molecular beam epitaxy. Full details on the semiconductor heterostructure, the experimental setup and device fabrication are included in the Methods Section and Supplementary Notes 1 and 2. The optical and SAW-properties of the as-fabricated devices were as follows: the optical waveguide losses of $(10.8 \pm 2.50)$ dB·cm$^{-1}$ of the fabricated IQPCs are competitive with the state of the art in this material system[34,35]. The insertion loss of the delay line is $S_{21} = 28$ dB at low temperatures. The full electromechanical conversion efficiency including losses at the cryostat wiring is 4%, proving efficient generation of the SAW on the comparably weak piezoelectric GaAs. Full details are included in Supplementary Notes 3.1 and 3.2. In the experiments presented in the remainder of the paper, SAWs are generated by a continuous wave (cw) rf signal(s) applied to one or two IDTs. We then assess the $\varphi_{SAW}$-dependence using a cw laser to photoexcite the QD and initialize the input qubit by the subsequentially emitted single photon. The output signals collected by the lensed fiber are analyzed in the time domain to obtain the $\varphi_{SAW}$-dependence. This

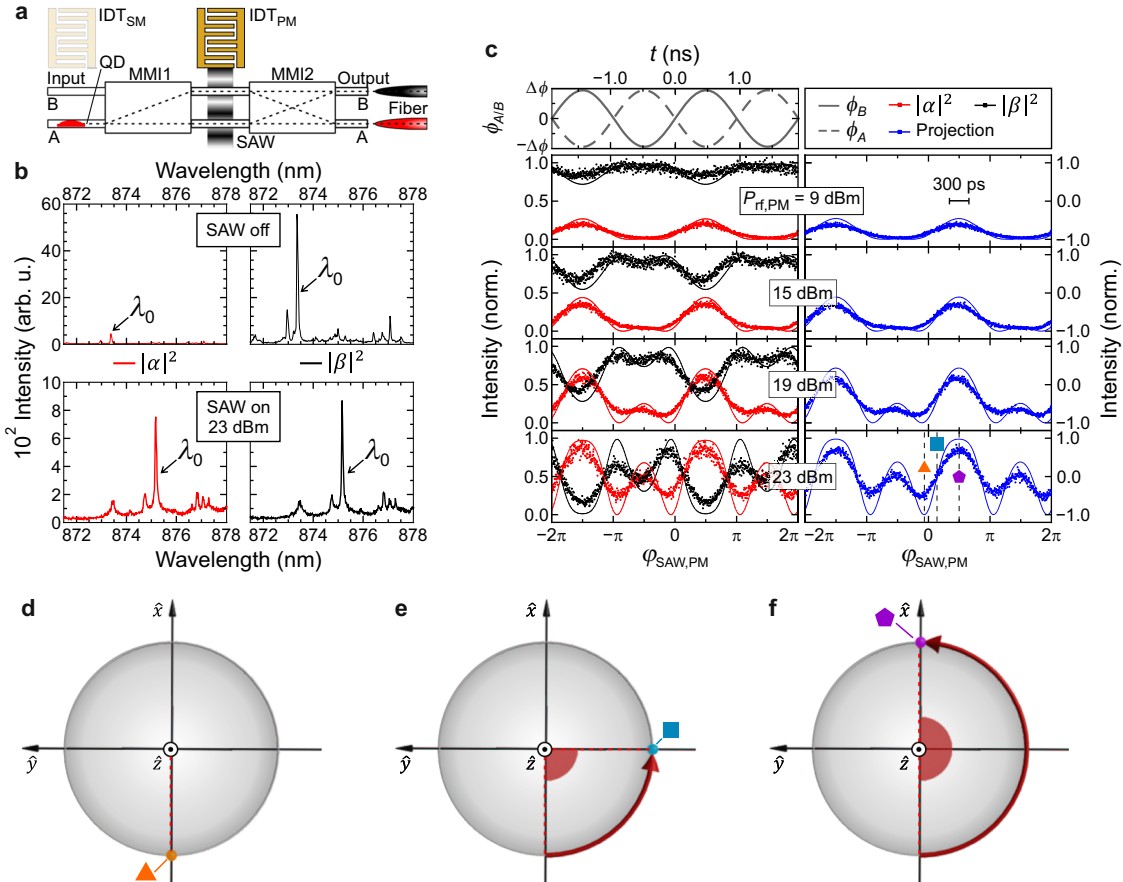

**Fig. 2 | Dynamic routing of quantum dot emission. a** Schematic representation of the experimental configuration: although the optical transition of a single QD in Input WG-A is not modulated (IDT$_{SM}$ is inactive), the response of the MZI is modulated (IDT$_{PM}$ is active) as the photons pass through. Emission is collected by a lensed fiber from either Output WG-A (red) or Output WG-B (black). **b** Phase-integrated emission spectra of the same single QD measured without a SAW (top panels, IDT$_{PM}$ off) and an applied SAW ($P_{rf,PM}$ = 23 dBm, lower panels). **c** Measured SAW phase-dependent intensity of the main QD emission line ($\lambda_0$ in **b**) detected via Output WG-A (red symbols) and Output WG-B (black symbols) for different $P_{rf,PM}$ as defined by Eq. (6) and Eq. (7). The simulation results (solid lines) for phase

modulation amplitudes $\Delta\phi$ = 0.3, 0.5, 0.8 and 1.3 rad are also shown for both outputs. The upper panel shows the time-resolved SAW-induced optical phase shift $\phi_{A/B}$ in the upper (WG-B, solid line) and lower arms (WG-A, dashed line) of the MZI. The right-hand side panel shows the $\langle S_Z \rangle$-projection of the qubit for the measurement and the corresponding simulation given by Eq. (8). The scale bar in the upper right panel shows the 300 ps timing jitter of the used single photon detector. **d**–**f** Rotations of the qubit states in the equatorial plane of the Bloch sphere at three distinct phases during the acoustic cycle given by Eq. (2) and Eq. (3). Symbols mark these phases in the projection data in (**c**).

scheme allows us to elegantly detect the dynamic modulation by the SAW. In our proof-of-principle study, the cw applied rf power leads to unwanted sample heating. The latter can be significantly suppressed by using a pulsed SAW excitation scheme[61] or in hybrid devices comprising a strong piezoelectrics e.g. LiNbO$_3$ with heterointegrated semiconductor QDs[92]. Full details on the experimental setup are presented in the "Methods" section and in Supplementary Note 2.

## Tuneable single photon routing and state rotation

We continue with the characterization of the dynamic modulation of the quantum interference in the MZI with a SAW generated by IDT$_{PM}$. The operation principle relies on the local modulation of the refractive index in the two MZI WGs and depends on the local phase of the SAW, $\varphi_{SAW,PM}$. The change in refractive index causes an optical phase shift $\phi(\varphi_{SAW,PM})$ between WG-A and WG-B and leads to a dynamic modulation of the interference at MMI2 and, thus, the superposition state of the output photonic qubit, given by Eq. (4)–Eq. (7). Figure 2a shows the schematic of the measurement configuration. A single QD is optically excited in Input WG-A to prepare a $|0\rangle$-like state of the photonic qubit at Input WG-A and WG-B.

First, we compare the phase-integrated photon output characteristics for the as-fabricated passive and active device in Fig. 2b. As

confirmed in the upper panel, the device cross-couples photons of the excited QD from Input WG-A to Output WG-B ($|\beta(\varphi_{SAW,PM})|^2$, right, black) and almost no emission is detected from Output WG-A ($|\alpha(\varphi_{SAW,PM})|^2$, left, red), as expected from Eq. (6) and Eq. (7) using $\phi$ = 0. This result corresponds precisely to the inversion of the input photonic qubit $|0\rangle \rightarrow |1\rangle$ with a high static fidelity or switching contrast of 0.94 ± 0.01. The small deviation arises from the fabrication-related imperfections' static phase $\phi_0$ = 0.51 rad. Supplementary Note 3.3 contains a full set of simulation data.

When an acoustic power of $P_{rf,PM}$ = 23 dBm and a frequency of $f_{SAW,PM}$ = 525 MHz is applied to IDT$_{PM}$ (lower panels), the signal is routed dynamically between both outputs. Thus, the phase-integrated spectra detected from Output WG-A (lower left panel, red) and Output WG-B (lower right panel, black) are almost perfectly identical. Note the spectra in the lower panels are shifted to longer wavelengths accompanied by a small reduction of the total emission intensity due to an increase of the temperature when the radio frequency signal is applied[93]. This is described in detail in Supplementary Note 3.6. For clarity, corresponding emission lines in the spectra are marked by $\lambda_0$. Importantly, no additional broadening is observed. Thus, no unwanted spectral modulation occurs when the quantum interference of the photonic qubit is dynamically controlled by $R_\phi(\varphi_{SAW,PM})$.

Second, we study the modulation of the qubit rotation induced by the $R_\phi(\varphi_{SAW,PM})$-gate as a function of $\varphi_{SAW,PM}$. The upper panel in Fig. 2c shows the time and phase dependence of the antiphased optical phase modulations, of $\phi_{A,B}$, in WG-A (dashed line) and WG-B (solid lines) of the MZI, respectively. In the left panels below, we analyze the intensity of the dominant QD emission line $\lambda_0 = 874$ nm collected from Output WG-A (red, $|\alpha(\varphi_{SAW,PM})|^2$) and Output WG-B (black, $|\beta(\varphi_{SAW,PM})|^2$) as a function of the acoustic phase, $\varphi_{SAW,PM}$, with the applied rf power increasing from top to bottom. At the lowest drive power $P_{rf,PM} = 9$ dBm the modulation of the refractive index and thus $\Delta\phi$ is weak. Even at this low power level however, the signals detected from the two outputs already exhibit the expected antiphased modulation. This indicates that the pure $|1\rangle$-output state of the unmodulated MZI is dynamically adopting $|0\rangle$-character due to the oscillation of $\phi(\varphi_{SAW,PM})$. When $P_{rf,PM}$ increases, the modulation amplitude of the optical phase, $\Delta\phi$, increases further and the resulting interference in the modulated MZI develops a pronounced oscillation. To better visualize the dynamic nature of the qubit rotation by the SAW, we extract the $Z$-projection of the qubit,

$$\langle S_Z \rangle = \frac{|\alpha(\varphi_{SAW,PM})|^2 - |\beta(\varphi_{SAW,PM})|^2}{|\alpha(\varphi_{SAW,PM})|^2 + |\beta(\varphi_{SAW,PM})|^2} \quad (8)$$

This figure of merit is plotted in the right panels. For low $P_{rf,PM}$, the qubit is predominantly in the $|1\rangle$ state, i.e $\langle S_Z \rangle = -1$. For the highest $P_{rf,PM} = 23$ dBm, the Bloch vector rotates between $\langle S_Z \rangle = -0.71$ and $\langle S_Z \rangle = +0.82$. From these experimental values we obtain the switching contrast of 0.75, which is competitive with electro-optic modulation in monolithic GaAs IQPCs with embedded QDs[52]. This value is a lower bound since the timing jitter of our detector of 300 ps (scale bar in Fig. 2c) limits the time resolution at the high modulation frequency $2f_{SAW,PM} = 1.05$ GHz of the data. A realistic value may range between this resolution limited value of 0.75 and the near-unity predicted by our simulation. Importantly, it is rotated into the equatorial plane ($\langle S_Z \rangle = 0$) at well-defined phases during the acoustic cycle. The non-zero static phase, $\phi_0$, gives rise to the observed device-characteristic beating of the dynamic qubit rotation while for a perfectly symmetric MZI ($\phi_0 = 0$) the frequency of this oscillation is given by $2f_{SAW,PM} = 1.05$ GHz. Note, that the non-zero $\phi_0$ does not represent a limitation of our concept because it can be adjusted after calibration if deemed necessary. A rigorous analytical model of this beating is detailed in Supplementary Note 3.4 together with the respective theoretical optical field intensity propagations.

Third, we validate our experimental findings by performing beam propagation method simulations considering $\phi(\varphi_{SAW,PM})$ in the MZI. Phase-dependent simulation results are indicated by the solid lines in Fig. 2c. They show that the experimental data can be nicely reproduced for $\Delta\phi$ ranging between $\Delta\phi = 0.3$ rad and 1.3 rad and near-unity $|\langle S_Z \rangle|$ can be predicted from the projection of the strongest modulation. Figure 2d–f then shows the rotation of the qubit states in the equatorial plane of the Bloch sphere at three distinct acoustic phases, $\varphi_{SAW,PM} = -0.06\pi, 0.14\pi$ and $0.5\pi$, which are accordingly marked in the projection measurement data in Fig. 2c. Clearly, the data presented in Fig. 2 validate that the applied SAW induces a dynamic $R_\phi(\varphi_{SAW,PM})$-gate and such driven qubit rotations enable the faithful generation of a superposition state in the equatorial plane of the Bloch sphere. Since these rotations are gated with $f_{SAW,PM} = 525$ MHz, the data prove dynamic routing of the on-chip generated single photons on sub-nanosecond timescales, which are challenging to reach in alternative electro-optic or nanoelectromechanical approaches[57,58].

## Conservation of single photon character

As the next step of our proof-of-principle study, we now prove that the single photon nature of the qubit is conserved. We measure the second-order correlation function $g^{(2)}(\tau)$ using a standard Hanbury-

Brown and Twiss setup and Supplementary Note 4 shows data of a reference QD with no modulation applied. These anti-bunching data in Supplementary Fig. 13 yield $g^{(2)}(0) = 0.48 \pm 0.03 < 0.5$. The observed values of $g^{(2)}(0)$ in our experiments are limited by the finite time resolution of our detectors and the non-resonant, above bandgap optical pumping and not by the SAW modulation and there exists no fundamental limitations[74,93,94] to reach the $g^{(2)}(0)$ record low levels reported for this platform[34,35]. Importantly, in the context of this paper, $g^{(2)}(0) > 0.5$ enables us to conduct proof-of-principle experiments and assess the impact of SAW modulation of our IQPC on the stream of single photons. In Fig. 3, we compare $g^{(2)}(\tau)$ of two different QDs with a frequency $f_{SAW,PM} = 525$ MHz SAW applied to IDT$_{PM}$ at a power level of $P_{rf,PM} = 17$ dBm. As shown by the schematics in the upper panels (Fig. 3a), the two QDs located in Input WG-A (Fig. 3a) and Input WG-B (Fig. 3b), initialize $|0\rangle$ and $|1\rangle$ input photonic qubit states, respectively. The time-dependent qubit rotations of both QDs are presented in Supplementary Note 4.2. Correlations are measured from Output WG-B, i.e. the time-dependent $|\beta(\varphi_{SAW,PM})|^2$ projection of the qubit on its $|1\rangle$ component. Data are fitted using a procedure provided in Supplementary Note 4.3. For the QD in Fig. 3a, the emitted single photon is routed to Output WG-B, i.e. $|0\rangle \rightarrow |1\rangle$ in the static case. At the applied power level, the modulation of the output intensities is weak [c.f. Fig. 2c]. Thus, $g^2(\tau)$ of this QD is expected to be like that of the unmodulated case with a weak $f_{SAW,PM}^{-1} = 1.9$ ns-periodic modulation superimposed. The measured $g^{(2)}(\tau)$ (blue line) shows precisely the expected anti-bunching behavior and $f_{SAW,PM}$ is clearly resolved in the fast Fourier Transform (FFT) of the data shown as an inset. From a best fit (red line) to the data considering the timing resolution of our detectors, we obtain $g^{(2)}(0) = 0.38 \pm 0.06$. For the QD in Fig. 3b, the emitted single photon is routed to Output WG-A, i.e $|1\rangle \rightarrow |0\rangle$ in the static case. At the selected SAW amplitude [c.f. Figure 2c], the time-dependent $|\beta(\varphi_{SAW,PM})|^2$ projection measured from Output WG-B is non-zero for short time intervals. Thus, the measured $g^2(\tau)$ exhibits a clear $f_{SAW,PM}^{-1} = 1.9$ ns-periodic modulation confirmed by the FFT of the data shown as an inset. Most importantly, the data (blue line) exhibit a clear suppression of coincidences at $\tau = 0$, with $g^{(2)}(0) = 0.42 \pm 0.1$ obtained from a best fit (red line). Both $g^{(2)}(0)$ values agree well with that of the unmodulated QD shown in Supplementary Note 4.1. Our proof-of-principle experiments unambiguously show that the anti-bunched single photon nature of the transmitted light and thus photonic qubit is preserved with the dynamic $R_\phi(\varphi_{SAW,PM})$-gate applied, which is a key requirement for practical applications.

## Dynamic wavelength-selective single photon multiplexing

Finally, we apply spectral modulation to the QD. This enables us to implement a proof-of-principle multiplexing and demultiplexing of single photons. To this end, we simultaneously modulate the spectral emission characteristics of the QD using IDT$_{SM}$ and dynamically control the qubit in the MZI using IDT$_{PM}$. A schematic of the experimental configuration is shown in Fig. 4a. We study a QD located in Input WG-B initializing a $|1\rangle$-input qubit and read-out is performed at Output WG-B ($|\beta|^2$). Both SAWs are active as we set $f_{SAW} = f_{SAW,SM} = f_{SAW,PM} = 524.12$ MHz and $P_{rf,SM} = P_{rf,PM} = 19$ dBm. Importantly, we can program the relative phase $\Delta\varphi_{SAW} = \varphi_{SAW,SM} - \varphi_{SAW,PM}$ between the SAWs driving the spectral and phase modulations, simply by setting the phases of the driving electrical signals.

To start, we confirm the spectral modulation of the quantum emitter. The measured phase-dependent emission spectra of the QD are shown in false-color representation as a function acoustic phase, $\varphi_{SAW,SM}$, and optical wavelength in the upper panel of Fig. 4b. The upper axis is in units of time during the SAW-cycle. The emission exhibits the expected sinusoidal modulation $\lambda(t) = \lambda_0 + \Delta\lambda \cdot \sin 2\pi f_{SAW}t$ due to the SAW-induced modulation through the deformation potential coupling[95]. For our devices we find a total tuning bandwidth of $2\Delta\lambda = 0.8$ nm (cf. Supplementary Fig. 10) comparable to recent reports of

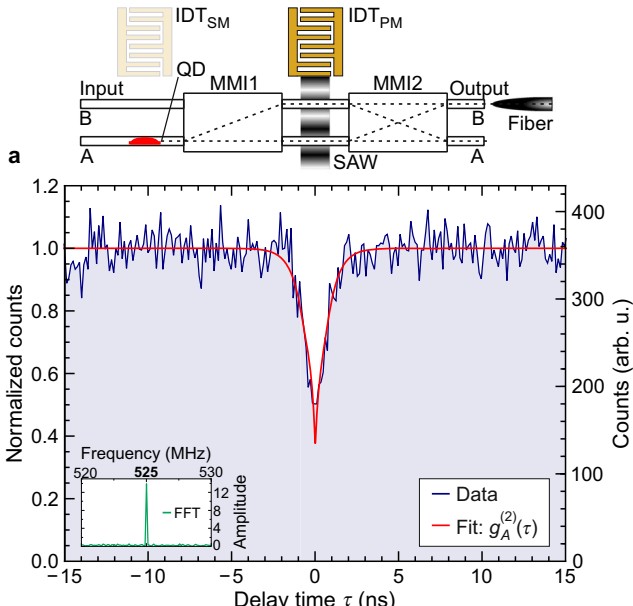

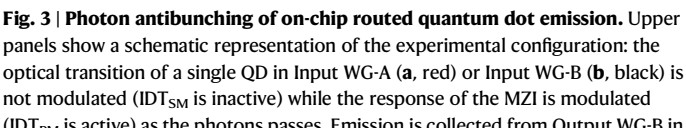

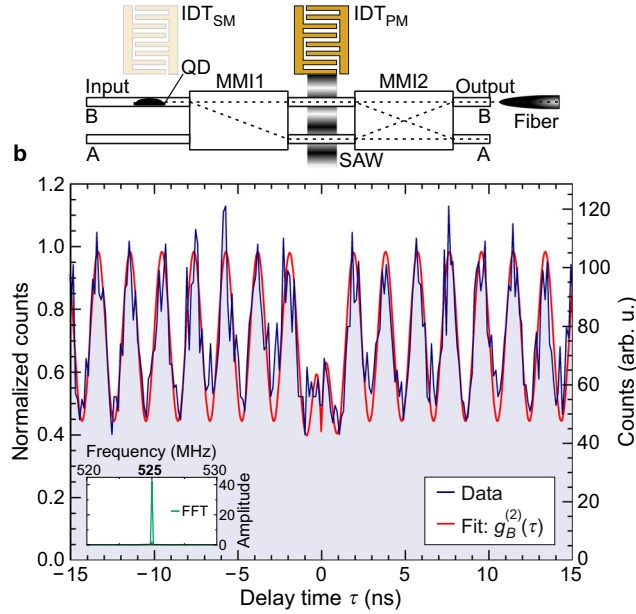

**Fig. 3 | Photon antibunching of on-chip routed quantum dot emission.** Upper panels show a schematic representation of the experimental configuration: the optical transition of a single QD in Input WG-A (**a**, red) or Input WG-B (**b**, black) is not modulated ($IDT_{SM}$ is inactive) while the response of the MZI is modulated ($IDT_{PM}$ is active) as the photons passes. Emission is collected from Output WG-B in both cases. Main panels show the measured second order correlation function $g^{(2)}$ ($\tau$) (blue) for the QD in Input WG-A (**a**) and Input WG-B (**b**) for $P_{SAW,PM}$ = 17 dBm and $f_{SAW,PM}$ = 525 MHz. The red lines are best fits to the data yielding: $g^{(2)}$ (0) = 0.38 ± 0.06 and $g^{(2)}$ (0) = 0.42 ± 0.09 in (**a**) and (**b**), respectively. Insets show the fast Fourier transform (FFT) of the data with a clear signal at $f_{SAW,PM}$.

static Stark tuning on this platform[81]. A detailed analysis is included in Supplementary Note 3.5. In the experiments presented in the following, we set $\Delta\lambda$ = 0.05 nm. The data clearly proves that a single photon is emitted at a well-defined wavelength at a given phase during the acoustic cycle. Since the QD is optically pumped by a continuous wave laser, all wavelengths are injected into the MZI and are thus, multiplexed at the respective phases. This spectral oscillation sets the reference for the MZI modulation driven by the SAW generated by $IDT_{PM}$. Next, the lower panel of Fig. 4b shows the dynamic single photon routing (QD in Input WG-B, i.e. $|1\rangle$ initialization of the qubit) as a function of phase from Output WG-B ($|\beta(\varphi_{SAW,SM})|^2$) for four different $\Delta\varphi_{SAW}$ = 0, $\pi/2$, $\pi$ and $3\pi/2$. The vertical lines connect to the emission wavelengths of the QD when the dynamically tuned quantum interference projects the input $|1\rangle$ -state to the output $|1\rangle$ -state. Clearly, $\Delta\varphi_{SAW}$ programs the projected wavelength: the center wavelength ($\lambda_0$ = 876.35 nm) is detected from Output WG-B for $\Delta\varphi_{SAW}$ = 0 (purple) and $\Delta\varphi_{SAW}$ = $\pi$(orange), while for $\Delta\varphi_{SAW}$ = 0.5$\pi$ (green), and for $\Delta\varphi_{SAW}$ = 1.5$\pi$ (blue), the maximum ($\lambda_{max}$ = 876.4 nm) and minimum wavelength ($\lambda_{min}$ = 876.3 nm) leave via this output, respectively. This dynamic, single photon wavelength de-multiplexing is demonstrated in phase resolved experiments. In Fig. 4c-f, we show the phase-dependent emission of the QD. The symbols mark the above selected $\Delta\varphi_{SAW}$. First, we observe that emission is detected only at distinct phases of the acoustic cycle. This confirms that the SAW-modulated MZI is operated as a tuneable, dynamic signal router. Second, as $\Delta\varphi_{SAW}$ is tuned, the wavelength of the photons exiting via Output WG-B changes. At $\Delta\varphi_{SAW}$ = 0 (Fig. 4c) and $\Delta\varphi_{SAW}$ = $\pi$ (Fig. 4e) the QD emission is coupled out during the rising and falling edge of the spectral modulation, respectively. At $\Delta\varphi_{SAW}$ = 0.5$\pi$ (Fig. 4d), $\lambda_{max}$ is filtered, perfectly antiphased to the detection of $\lambda_{min}$ at $\Delta\varphi_{SAW}$ = 1.5$\pi$ (Fig. 4f). Lastly, we further corroborate the faithful demultiplexing in Fig. 4g. We plot the intensities at $\lambda_{max}$ (red), $\lambda_{min}$ (blue) and $\lambda_0$ (black) over two complete cycles of $\Delta\varphi_{SAW}$. These data unambiguously confirm that $\lambda_{max}$ and $\lambda_{min}$ are filtered at well-defined $\Delta\varphi_{SAW}$. Furthermore, the center wavelength ($\lambda_0$) oscillates at 2$f_{SAW}$ because the applied demultiplexing filters this wavelength at both the rising and falling edges of the spectral modulation.

## Discussion

In conclusion, we designed, monolithically fabricated, and demonstrated proof-of-principle operation of key functionalities important for hybrid photonic and phononic quantum technologies. We used integrated quantum emitters to generate single photons in an IQPC with acoustically tuneable wavelength. We executed dynamic and tuneable rotations in a compact SAW-modulated MZI which faithfully preserve the single photon nature. We show dynamic 2$f_{SAW} \approx$ 1.05 GHz routing of the QD-emitted single photons between the two outputs exceeding that of electro-optic and nanoelectromechanical tuning on this platform by more than a factor of 100. Moreover, our scheme enables the generation of output states perfectly located in the equatorial plane of the Bloch sphere providing a dynamically tuneable on-chip beamsplitter. Finally, we implemented spectral multiplexing of the emitted single photons by two SAWs, one dynamically straining the emitter which are demultiplexed by a variable acoustic phase-lock to the second SAW driving the single qubit rotation.

The reported proof-of-principle results open several exciting perspectives. First, full arbitrary unitary beamsplitter operation and thus single qubit rotations[91] are straightforward and can be realized simply by adding another IDT at the Output WGs to execute a second phase gate. Second, the purity of the single photon emission can be enhanced by embedding the QDs in SAW-tuneable photonic cavities[68,96]. When establishing stable phase lock between a train of excitation laser pulses and the SAW, the latter modulates the cavity-emitter coupling and precisely triggers the Purcell-enhanced emission of single photons[93]. Third, our approach is scalable because the low propagation loss of SAWs allows to modulate multiple photonic systems and QDs by a single SAW beam[97] for parallelized control schemes. Fourth, power levels required for switching can be significantly reduced further by embedding the photonic components in phononic resonators and waveguides to enhance the interactions[65,66,83,98–101]. Fifth, all demonstrated functionalities can be transferred to hybrid architectures. Important examples include the heterointegration on LiNbO$_3$ SAW and IQPC devices[46,92], with 100-fold enhanced electromechanical coupling compared to monolithic GaAs devices or CMOS

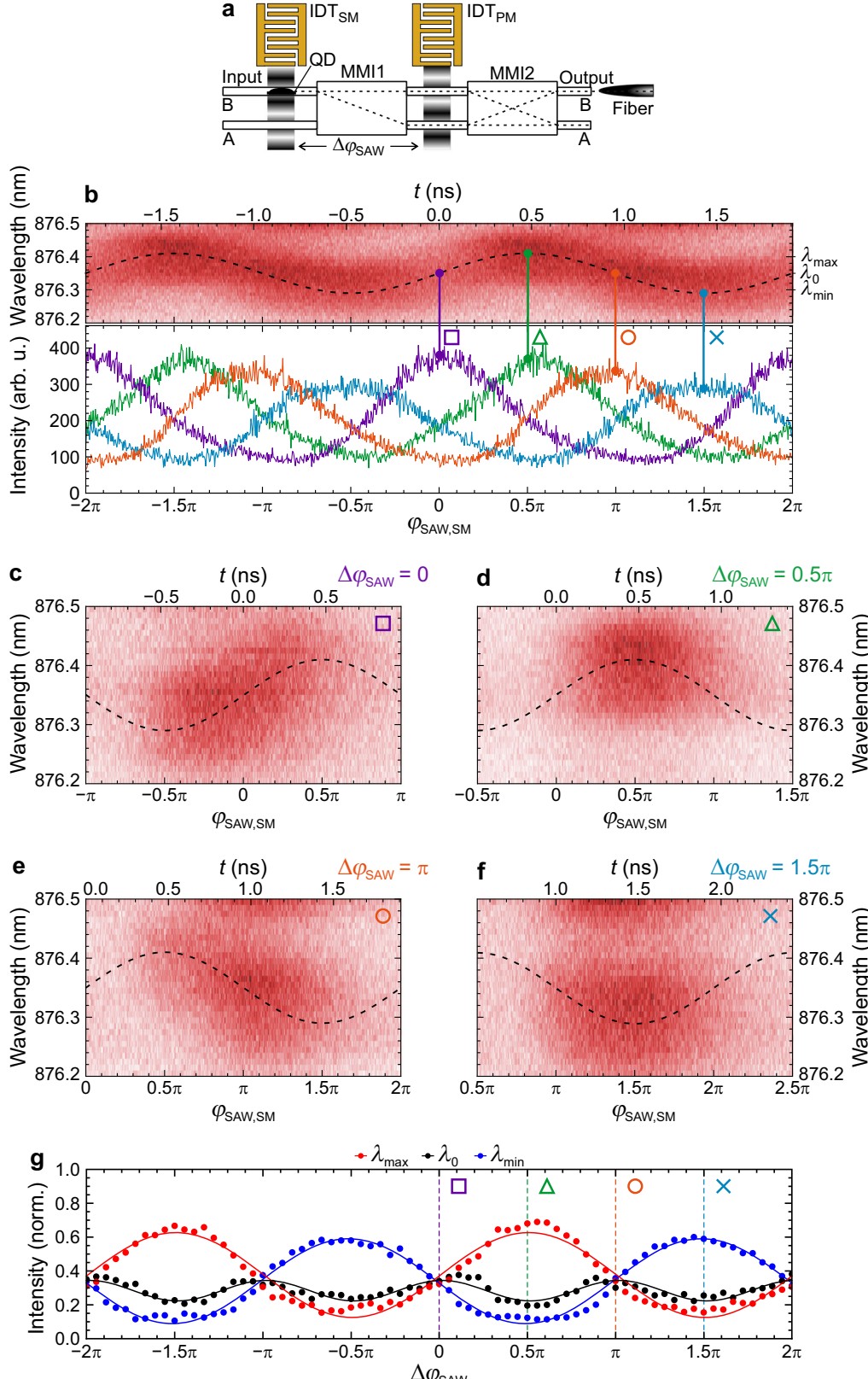

**Fig. 4 | Single photon (de)multiplexing. a** Schematic representation of the experimental configuration: Both IDTs are active. The optical transition of a single QD in Input WG-B is modulated by $IDT_{SM}$ while the response of the MZI is modulated by $IDT_{PM}$ as photons pass through. **b** Upper panel: Acoustic phase-dependent spectral modulation of the QD. Lower panel: intensity modulations at Output WG-B at four distinct relative phases $\Delta\varphi_{SAW}$. **c–f** Acoustic phase-dependent emission spectra of the QD measured from Output WG-B for the four distinct relative phases marked in (**b**). **g** Programmable spectral demultiplexing of the maximum ($\lambda_{max}$, red symbols), center ($\lambda_0$, black symbols) and minimum ($\lambda_{min}$, blue symbols) of the spectral modulation as a function of $\Delta\varphi_{SAW}$. Colored lines are best fits to the data.

compatible Silicon IPCs with AlN piezoelectric coupling layers[56] and heterogeneously integrated III-V QDs[39]

Additionally, we note that the presented concept can be directly applied to other types of quantum emitters[19,45,73,75,102,103] and spin degrees of freedom[104–107], even in the resolved sideband regime[71,74]. Moreover, they can be combined with static electric field[81,108] or stressors[109] tuning to dissimilar QDs could be tuned into resonance to create multi-qubit systems. The integration density can be increased for instance by implementing a phase gate in a single MMI[110] and multi-port MMIs enable phased-array wavelength-division multiplexing[111].

## Methods

### Sample design and fabrication
The heterostructure was grown by molecular beam epitaxy on a semi-insulating (001) GaAs substrate. It consists of a 1500 nm $Al_{0.2}Ga_{0.8}As$ cladding layer followed by a 300 nm thick GaAs waveguide layer with a single layer of optically active self-assembled (In,Ga)As QDs in its center. Devices were fabricated monolithically on these substrates using optical lithography. The IQPCs were etched using inductively coupled plasma reactive ion etching ICP-RIE. IDTs (10 nm Ti/30 nm Al/ 10 nm Ti) were fabricated by a standard lift-off process.

### Device simulation
MMI dimensions were calculated using established numerical simulation methodologies[112]. The full IQPC comprising individual MMI dimensions, tapered waveguides, S-bends and the respective waveguide interfaces was optimized by finite difference 3D beam propagation method simulations using commercial software packages. For the simulation of the active device a sinusoidal modulation of the refractive index in the optomechanical interaction region of the MZI arms was applied. Device parameters and full details on the model used for the modulated transmission behavior can be found in Supplementary Note 1 and 3.

### Experimental setup
Experiments were conducted in a cryogenic photonic probe station which is equipped with custom-made radio frequency lines. A schematic of the full setup and a detailed description are included in Supplementary Note 2. In essence, QDs were optically excited by a non-resonant continuous wave laser ($\lambda_{laser}$ = 660 nm) under normal incidence. The QD emission is collected from the cleaved end facets using a lensed fiber and spectrally filtered by a 0.75 m grating monochromator. A cooled CCD detector or up to two silicon single photon detectors (timing resolution 300 ps) are used for time-integrated and time-resolved detection, respectively. The outputs of up to two locked radio frequency signal generators are applied to the IDTs for SAW generation. Additionally, time-resolved optical detection was referenced to these electrical signals[93,113,114].

## Data availability
The raw data generated in this study are available in the ZENODO database at https://doi.org/10.5281/zenodo.7298560.

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

## Acknowledgements

This work has received funding from the European Union's Horizon 2020 research and innovation programme under the Marie Skłodowska-Curie grant agreement No. 642688 (SAWtrain) and the Deutsche Forschungsgemeinschaft (DFG, German Research Foundation) via the Cluster of Excellence "Nanosystems Initiative Munich" (NIM). K.M. acknowledges financial support via the German Federal Ministry of Education and Research (BMBF) via the funding program Photonics Research Germany (contract number 13N14846). J.J.F. and K.M. acknowledge financial support by the Deutsche Forschungsgemeinschaft (DFG, German Research Foundation) under Germany's Excellence Strategy—EXC-2111—390814868. We thank W. Seidel and S. Rauwerdink (PDI) for cleanroom device processing, and Hubert Riedl (WSI-TUM) for crystal growth. We thank Achim Wixforth and Andres Cantarero for enduring and continuous support and fruitful discussions.

## Author contributions

H.J.K. conceived and designed study and coordinated project. D.D.B. and M.W. coordinated the experimental work. D.D.B. designed device supported by M.M.d.L., A.C.P., M.W., and E.D.S.N. M.W. conducted experiments. D.D.B. and M.W. analyzed data. A.C.-P. and P.V.S.

fabricated device. K.M. and J.J.F. provided the semiconductor hetero-structure. H.J.K. and M.M.d.L. supervised the project. H.J.K. and D.D.B. wrote the manuscript.

## Funding

## Competing interests

The authors declare no competing interests.
