## [Peer Review File · Nature Communications]

On-chip generation and dynamic piezo-optomechanical rotation of single photonsEditorial Note: This manuscript has been previously reviewed at another journal that is not operating a transparent peer review scheme. This document only contains reviewer comments and rebuttal letters for versions considered at *Nature Communications*.

REVIEWER COMMENTS

Reviewer #1 (Remarks to the Author):

I find that the authors have addressed all of my comments sufficiently well in the revised manuscript, and I have no new concerns. I think the manuscript does a good job highlighting the promise of the presented platform, and potential pitfalls. I am happy to recommend publication as is in *Nature Communications*.

Reviewer #2 (Remarks to the Author):

The authors have reported an integration of a single QD single-photon source and a surface acoustic wave (SAW)-driven MZI in GaAs, and their statistic measurement and SAW dynamic control of single-photon state. The authors have partially addressed my comments and concerns. Figure 1b is still misleading (a photonic qubit cannot be presented by the positions of QD). The matrix representation of MMI and MZI are used now instead of incorrect operator representations in the original version (in fact I meant their correct operator representations read much better). As I have pointed out, low-fidelity and “random”-response of the SAW phase-gate are shown in Fig.2, as well as strong crosstalk. The authors should have discussed how to address these technological issues for large-scale implementation for quantum optical applications. The authors pointed out the fidelity of a specific Pauli-X ($|\text{ket}\{0\}\rangle$ to $|\text{ket}\{1\}\rangle$ is 0.94) but they hidden the values for other operations (the whole interference fringe). The authors should have fairly compared the SAW and EO (cryogenics) phase modulators. My main concerns when considering its publication in high-reputation journals like *Nature Photonics* and *Nature Communications* are the following (I have pointed out before): in order to demonstrate the technological advance of a new technology, here it is the SAW control of quantum state of light, we better to know the what could be possible limitations and how to address them. I do not agree that these discussions are out of the scope of a high-quality work even it is a proof-of-principle demonstration. I would suggest the authors clarify these points. With that, I think the advance of the SAW technology could be much more acknowledged. Finally, I would strongly suggest the authors highlight all modified text and figures in the revised manuscript in future.

Reviewer #3 (Remarks to the Author):

The authors demonstrated the on-chip initialization, rotation and dynamic routing of a photonic qubit. They have used on-chip generated surface acoustic waves to imprint a time-dependent optical phase in an interferometer to modulate the qubit rotation and to spectrally tune the quantum dot emitters. The different elements are integrated on a III-V semiconductor chip. The paper is a technically impressive extension (integration of phase rotation, MZI) of the previous work by the authors and other groups towards integrated quantum photonics chips.

I appreciate that the main purpose of the presented work is a “proof-of-principle study” as stated by the authors. The results are presented and analysed in detail in a proper way and some improvements recommended by the referees have been performed with respect to the previous version which has been submitted to *Nature Photonics*. The manuscript should be definitely published in some form.

However, I think attracting a broader readership requires also that the achieved performances are good or that the presented concepts are new or both. But this is not the case for the presented work due to some severe shortcomings. For example, the $g_2(0)$ values are not good, the contrast in wavelength switching is not very convincing and the unwanted local increase in temperature prevents the generation of indistinguishable photons and scalability. In my opinion, it is not enough to mention that this can be overcome by using pulsed SAW, better QDs etc., and giving reference to own previous work. For example, why has pulsed SAW not been performed in this study?

Furthermore, most of the presented concepts have already been shown in previous work. The tuning of QD emission wavelength, the fabrication of ridge waveguides and interdigital transducers have already been demonstrated by the authors individually in previous works.

Therefore, the authors have not convinced me on the potential scalability and/or novelty of their approach. At the current stage, I cannot recommend publication in Nature Communication.

Rebuttal

We thank the reviewers for their insightful comments and stating that our work deserves publication. We address their constructive and helpful comments point-by-point in the following.

Comments are consecutively numbered and printed in **red font**. Our replies / changes made are in **green font**. All changes are highlighted in **yellow** in the marked copy.

In this rebuttal we describe the most prominent and important changes made. In addition to these major changes, several smaller amendments and additions are made. All changes, major and minor are highlighted in the marked copy, however.

Reviewer #1 (Remarks to the Author):

I find that the authors have addresses all of my comments sufficiently well in the revised manuscript, and I have no new concerns. I think the manuscript does a good job highlighting the promise of the presented platform, and potential pitfalls. I am happy to recommend publication as is in Nature Communications.

We thank the reviewer for recognizing the significance of our work and recommending our paper for publication.

Reviewer #2 (Remarks to the Author):

The authors have reported an integration of a single QD single-photon source and a surface acoustic wave (SAW)-driven MZI in GaAs, and their statistic measurement and SAW dynamic control of single-photon state. The authors have partially addressed my comments and concerns.

We thank the reviewer for his/her insightful comments and pointing out points which need further improvement before publication.

Comment 2.1

Figure 1b is still misleading (a photonic qubit cannot be presented by the positions of QD). The matrix representation of MMI and MZI are used now instead of incorrect operator representations in the original version (in fact I meant their correct operator representations read much better).

Reply 2.1:

The reviewer is correct that a fully-fledged initialization of a qubit is not possible by choosing two different QDs in different arms.

Changes made:

In the revised paper, we do not refer to photonic qubit initialization anymore and use the term single photon generation:

- **On page 4, lines 128-130 for instance read now:**

As explained above the QDs emit single photons in the input waveguides WG-A and WG-B. Since the QDs are located in different WGs, our single photon generation scheme does not strictly correspond to a fully-fledged initialization of a photonic qubit. Nevertheless, we apply the corresponding terminology in the following because the state control scheme is equivalent to a qubit rotation.

- Figure are corrected accordingly:

Panel **b** is labelled “Single Photon Generation”

- The title was changed to

“On-chip generation and dynamic piezo-optomechanical rotation of single photons”
avoiding the term photonic qubit.

- Our scheme and the different gate operations are now introduced in more detail to make the paper more accessible for the diverse readership of Nature Communications. These changes are highlighted starting line 138, page 4 to line 203, page 6.

Comment 2.2

As I have pointed out, low-fidelity and “random”-response of the SAW phase-gate are shown in Fig.2, as well as strong crosstalk. The authors should have discussed how to address these technological issues for large-scale implementation for quantum optical applications. The authors pointed out the fidelity of a specific Pauli-X ($|\text{ket}\{0\}$ to $|\text{ket}\{1\}$) is 0.94) but they hidden the values for other operations (the whole interference fringe).

Reply 2.2

The reviewer is correct that the analysis and discussion of the data in Figure 2 **c** has to be improved. From the data we obtain a switching contrast of 0.75, which is competitive with electro-optic modulation (Midolo et al. 2017). We note that our fast $f_{mod} = 2f_{SAW,PM} = 1.05$ GHz modulation has to be compared to the timing jitter of our Si single photon detectors. The latter is 300 ps compared to $\frac{1}{f_{mod}} = 238$ ps. Thus, e.g. the switching to $\langle S_Z \rangle = -1$ marked by red triangle in Figure 2 **c** is not fully resolved and the real value is underestimated. The SAW response of the phase gate is not “random” but in good agreement with our simulation. We assume that the reviewer refers to the finite static optical phase offset which is related to imperfections in the nanofabrication. This static phase can be eliminated by proper calibration as performed in the paper and if deemed necessary in future work by post fabrication tuning.

Change made:

The revised paper contains now a more detailed analysis and discussion:

- For example lines 291-295 on page 9 now read:

From these experimental values we obtain the switching contrast of 0.75, which is competitive with electro-optic modulation in monolithic GaAs IQPCs with embedded QDs⁵². This value is a lower bound since the timing jitter of our detector of 300 ps (scale bar in Figure 2c) limits the time resolution at the high modulation frequency

$2f_{\text{SAW,PM}} = 1.05$ GHz of the data. A realistic value may range between this resolution limited value of 0.75 and the near-unity predicted by our simulation.

- Figure 2:

We included a scalebar showing the timing resolution in panel c. The caption has been modified accordingly.

Comment 2.3

The authors should have fairly compared the SAW and EO (cryogenics) phase modulators.

Reply 2.3:

We agree that electro-optic (EO) modulation represents an alternative route for phase modulation. The same holds for nanomechanical tuning in suspended devices. To our knowledge, the state of the art for monolithic GaAs devices with integrated QDs like ours are (Midolo et al. 2017) for EO tuning and (Papon et al. 2019) NEMS-based nanomechanical tuning. These works are excellent benchmarks because both report on single photon routing as we do. The main characteristics of these works and our works are compared in Table 1.

Table 1 – Comparison of main characteristics of implemented EO tuning and NEMS tuning and our work.

	EO Tuning Midolo et al.	NEMS Tuning Papon et al.	This work
Device layout	Tunable interference (Y-Splitter – modulated waveguide – beamsplitter)	Tuneable Beamsplitter only	Tuneable Mach Zehnder (Beamsplitter – modulated waveguide – beamsplitter)
Dimensions	400 μm modulated segment length Total device size not specified, estimated 300 μm x 300 μm	30 μm x 5 μm (Beamsplitter)	100 μm modulated section length; 3300 μm full device (incl. spectral modulator) x 40 μm width
Demonstrated Bandwidth	2.8 \pm 0.5 MHz (broadband, 3dB cut-off)	1.35 MHz (mechanical mode, resonant)	> 1GHz (quasi-resonant, determined by transducer design)
Resolved sideband regime accessible	No	No	Yes
Parallelization	Limited, each element requires individual contacts	Limited, each element requires individual contacts	Yes , multiple devices can be controlled by a single remote IDT / SAW beam
Operation voltage	-1 V to +1.5 V	Up to 10 V	23 dBm, 3.16 V(rms), 4.46 V(peak)

State rotation	Yes	No	Yes
Contrast	66%	99.5%	75%
Photon anti-bunching	No	Yes	Yes
Spectral modulation QD	no	no	Yes
Intrinsic losses of mechanism	Yes, (Franz-Keldysh) electro-absorption	No	No

The main results to compare to ours are summarized in Figure 1 and Figure 2.

Dynamics

Figure 1 – Comparison of EO and NEMS controlled single photon routing dynamics

Figure 1 show the dynamics of EO tuning (left) and NEMS tuning (right). The operation of both devices is limited to a few MHz. In contrast, SAWs can be inherently fast. We demonstrate single photon routing operation at frequencies exceeding 500 MHz, at least 100 times larger than EO and NEMS tuning. This marks a major and significant advancement. Such high frequencies are imperative for resolved sideband modulation (Matthias Weiß et al. 2021).

Switching contrast

Figure 2 – Comparison of EO and NEMS single photon routing contrast

Figure 2 shows the switching contrast for EO (left) and NEMS tuning (right). The switching contrast in our SAW-driven devices is at least on par with EO modulation. The contrast reported for NEMS tuning still outperforms ours, but similar levels have been reached by us in passive devices.

We note the dynamic nature of the SAW-technique renders a direct comparison of the switching contrast difficult. At the highest modulation contrast modulations occur at $2f_{SAW} > 1$ GHz. The time corresponding to $\frac{T_{SAW}}{2 \cdot 4} < 250$ ps is less than the timing jitter of our detectors (300 ps). Thus, the switching contrast defined as $|I_{max}| + |I_{min}|/2$ extracted from the experimental data is 0.75. Using faster detectors, e.g. superconducting nanowire single photon detectors with timing jitter in the 20-30 ps range would overcome this limitation. Unfortunately, this type of detector is currently not available in our laboratory. Nevertheless, the value deduced directly from our data is – as a lower bound – competitive with EO switching and the real value would range between this lower bound and near unity predicted by our simulation.

Scalability

Finally, we address the potential of scalability. For EO and NEMS modulation each modulator requires individual contacts since the modulation voltages have to be applied directly at the active region. Figure 3 shows a micrograph of one of the devices fabricated for this work. In the micrograph, three IDTs are visible. The IDT on the left is used for spectral modulation of the QDs. The two IDTs on the right can be used to modulate the phase of an MZI. Since SAWs propagate with low dissipation, multiple devices can be addressed by a single SAW beam. In the image shown, five WG devices are positioned in the propagation path (green and red) facilitating dynamic spectral modulation of the QD and MZIs (MMIs highlighted).

This scheme can be extended even further to multi-arm (waveguide grating) MZI-type devices which have been demonstrated with off-chip laser sources at room temperature (Crespo-Poveda et al. 2015). There is no fundamental limitation that these schemes could be implemented with on-chip quantum emitters for advanced multiplexing applications.

Figure 3 – Microscope image of sample: multiple waveguide devices can be modulated by a single SAW beam.

On chip dynamic tuning of QDs

In context of this comment, we would like to set emphasis on the spectral tuning of QDs in photonic structures. In this work we demonstrate tuning amplitudes ≥ 0.4 nm, which is on par with the state of the art on ridge waveguide devices using quantum confined Stark effect (QCSE) tuning reported by (Schnauber et al. 2021). We note that for suspended structures as used in the works by (Midolo et al. 2017) and (Papon et al. 2019), SAWs, or precisely Lamb waves enhance the tuning amplitude by more than a factor of 10 (Vogele et al. 2020).

The modulation frequency of approx. 0.5 GHz in our work is not limited by the scheme itself. Here state of the art is 3.6 GHz in SAW resonators (Imany et al. 2022) and 1.7 GHz for propagating SAWs (Matthias Weiß et al. 2021). Even for the demonstrated 0.5 GHz modulation, spectral multiplexing of the QD-emitted photons in the resolved sideband regime is feasible (Wigger et al. 2021).

As pointed out above, a single SAW beam can modulate literally any QD in its propagation path. For radio frequency QCSE tuning requires small diode footprints to reduce the capacitance and, thus RC time constant of the device (Pagliano et al. 2014; Widhalm et al. 2018; Widhalm et al. 2021). This in turn typically requires small diode footprints, individual contacting schemes and elaborate impedance matching. For SAWs, these complications are alleviated since a single transducer can be employed which requires one ground and one signal wire and impedance match of that.

Change made:

We included a comparison to two alternative tuning mechanisms in the manuscript which now highlight better the points of novelty:

- The last paragraph of the introduction (page 2-3, lines 77-90) now reads:

Our demonstrated operation bandwidth of > 1 GHz for single photon routing exceeds that of reported for state-of-the-art monolithic devices employing electro-optic⁵² and nanomechanical⁵⁸ tuning by more than a factor 100. The underlying mechanical tuning mechanism does not induce inherent losses, in contrast to the well-known Franz-Keldysh electroabsorption in electro-optic devices⁷⁷. Also, our achieved spectral tuning range of the integrated QD of ≥ 0.8 nm is competitive with Stark-effect tuning on this platform⁷⁸. The achieved > 1 GHz operation nests our system well in the resolved sideband regime enabling on-chip parametric quantum phase modulation of the QD^{71,74,79,80} and the routed single photons⁸¹. The electrical generation of SAW and their ultralow dissipation offers distinct advantages over local tuning schemes. Thermo-optic, electro-optic or Stark-effect tuning require local electrodes to generate heat or electric fields for each element. SAWs, in contrast however can be piezo-electrically generated by applying a rf voltage to an interdigital transducer (IDT) and the propagating SAW beam modulates any IQPC element or QD in its propagation path. These unique properties together with the ability to synchronize SAWs and optical pulses⁸² pave the way towards parallelized control in large scale IQPC networks.

- Another example is that parallelized addressing of multiple IQPCs which is discussed on page 14, lines 435-437:

Third, our approach is scalable because the low propagation loss of SAWs allows to modulated multiple photonic systems and QDs by a single SAW beam⁹⁵ for parallelized control schemes.

- As mentioned in Reply 2.2, we included a detailed discussion of Figure 2:

Figure 2 now also shows a scale bar indicating the timing resolution to corroborate the fact that the switching contrast is limited by the detector.

Comment 2.4

My main concerns when considering its publication in high-reputation journals like Nature Photonics and Nature Communications are the following (I have pointed out before): in order to demonstrate the technological advance of a new technology, here it is the SAW control of quantum state of light, we better to know the what could be possible limitations and how to address them. I do not agree that these discussions are out of the scope of a high-quality work even it is a proof-of-principle demonstration. I would suggest the authors clarify these points. With that, I think the advance of the SAW technology could be much more acknowledged.

Reply 2.4:

Thank you for this point which is related to the previous Comment 2.3. SAW control of integrated photonic circuits is indeed an emerging field and advantages have been pointed out in high-profile publications recently e.g. (Kittlaus et al. 2021; Shao et al. 2019; Sarabalis et al. 2020; Tadesse and Li 2014). So far, passive systems i.e. without integrated sources have been studied on different material platforms which are either strong piezoelectrics (Shao et al. 2019; Sarabalis et al. 2020; Tadesse and Li 2014) or highly advanced silicon IPCs with piezoelectric coupling layers for SAW generation (Kittlaus et al. 2021). The advantages pointed out in these and other important work also hold for our system. In fact, the advanced material platforms used there, are well suited for heterointegration of quantum emitters. Heterointegration overcomes the limitation of the presented device by boosting the conversion efficiency of the driving electrical signals to SAW.

Change made:

We revised the manuscript to highlight the advantages and prospects of SAWs.

- In the perspective paragraph (page 14, lines 429-443) several changes have been made.

For example the last sentence on heterointegration now reads:

Important examples include the heterointegration on LiNbO₃ SAW and IQPC devices^{46,90}, with 100-fold enhanced electromechanical coupling compared to monolithic GaAs devices or CMOS compatible Silicon IPCs with AlN piezoelectric coupling layers⁵⁶ and heterogeneously integrated III-V QDs³⁹.

Parallel addressing of multiple IQPCs by a single SAW-beam is explicitly mentioned:

Third, our approach is scalable because the low propagation loss of SAWs allows to modulated multiple photonic systems and QDs by a single SAW beam⁹⁵ for parallelized control schemes.

Finally, I would strongly suggest the authors highlight all modified text and figures in the revised manuscript in future.

Thank you for this recommendation. We submit a marked copy as well. We refrained submitting a marked copy of the transferred paper because it was hard to read.

Reviewer #3 (Remarks to the Author):

The authors demonstrated the on-chip initialization, rotation and dynamic routing of a photonic qubit. They have used on-chip generated surface acoustic waves to imprint a time-dependent optical phase in an interferometer to modulate the qubit rotation and to spectrally tune the quantum dot emitters. The different elements are integrated on a III-V semiconductor chip. The paper is a technically impressive extension (integration of phase rotation, MZI) of the previous work by the authors and other groups towards integrated quantum photonics chips.

I appreciate that the main purpose of the presented work is a “proof-of-principle study” as stated by the authors. The results are presented and analysed in detail in a proper way and some improvements recommended by the referees have been performed with respect to the previous version which has been submitted to Nature Photonics. The manuscript should be definitely published in some form.

We thank the reviewer for the favorable assessment and that the reviewer would like to see this work being published

Comment 3.1:

However, I think attracting a broader readership requires also that the achieved performances are good or that the presented concepts are new or both. But this is not the case for the presented work due to some severe shortcomings. For example, the $g_2(0)$ values are not good, the contrast in wavelength switching is not very convincing and the unwanted local increase in temperature prevents the generation of indistinguishable photons and scalability.

In my opinion, it is not enough to mention that this can be overcome by using pulsed SAW, better QDs etc., and giving reference to own previous work. For example, why has pulsed SAW not been performed in this study?

Reply 3.1:

We agree with the reviewer, that there is potential to improve the performance of this class of devices. However, we are convinced, that in this first decisive step, which we made in our opinion in this work, a full-scale optimization is not imperative. We do not claim that we have realized an application-ready or even market-ready device and are convinced that conducting meaningful characterization experiments are more than sufficient at this level.

We politely disagree that the work is lacking points of novelty. As we will show in Reply 3.2 to the following Comment 3.2, that the demonstrated performance indeed goes far beyond the state of the art in terms of modulation frequency. Also, the presumed low switching contrast is on par with EO modulation schemes as pointed out in Reply 2.3.

We explicitly answer the reviewer's question why no pulsed SAW experiments were performed: the data obtained in this type of experiment convey the same information on the dynamic switching than the performed cw-SAW experiments. However, their interpretation and, thus presentation in a paper are significantly more difficult which reduces the accessibility for non-expert readers.

We also consider the point that referring to own previous work invalidating the presented results not fair. This indirectly casts doubt on our expertise demonstrated over many years in numerous high-profile publications. Based on our experience these works unambiguously show that the limitations can be overcome. The reviewer indirectly states that the work is lacking scientific rigor. If we would have any doubts that we are missing evidence, appropriate supporting experiments would have been conducted or a new generation of samples would have been fabricated.

The reviewer is completely right, that in the next generation of devices and experiments a series of new key questions must be addressed. These include benchmarking the indistinguishability, interferometric detection and arbitrary qubit rotations using additional SAW-tunable MZIs, proper qubit initialization by the same QD (see Comment 2.1) and parallelization (modulating several QIPCs by the same SAW beam) to confirm scalability also on this particular SAW-MZI platform.

Comment 3.2:

Furthermore, most of the presented concepts have already been shown in previous work. The tuning of QD emission wavelength, the fabrication of ridge waveguides and interdigital transducers have already been demonstrated by the authors individually in previous works.

Reply 3.2:

We thank the reviewer for this comment. Several of these points are addressed in detail in Reply 2.3. Here we again summarize the key advances of our work beyond the state of the art.

The reviewer is correct that the reported work successfully combines two concepts to synergistically harness the individual functionalities. We point out that albeit the individual concepts may be the state-of-the-art in the SAW-field, their combination marks a major advancement, and the obtained results outperform or are to the best of our knowledge at least on par with or significantly outperform the state of the art of tunable quantum integrated photonic circuits with integrated emitters both on monolithic or hybrid architectures where generation and manipulation are on the same chip (i.e. not taking the single photons from a (high-performance) QD single photon sources and injecting these single photons into the photonic chip.

This is best evident when comparing to monolithic III-V QIPCs:

- All reports on waveguide based monolithic devices are quasi-static (Reithmaier et al. 2013; Reithmaier et al. 2015; Schnauber et al. 2018; Schnauber et al. 2021) – we show switching at a base frequency of 0.5 GHz (optical modulation at 2x0.5 GHz!)
- Similar holds for hybrid platforms with III-V quantum emitters (Zadeh et al. 2016; Elshaari et al. 2017; Elshaari et al. 2018; Davanco et al. 2017; Aghaeimeibodi et al. 2018) and other

types of quantum emitters (Errando-Herranz et al. 2021; Wan et al. 2020; Tonndorf et al. 2017).

- Dynamic control of QDs in ridge Waveguide-IPCs using the quantum confined Stark effect (QCSE) is reported by (Schnauber et al. 2021). The reported modulation amplitude is on par with our work.
- Reports on dynamic control of emitters in emitter-cavity coupling beyond 100 MHz are scarce and include our own work (M. Weiß et al. 2016) employing SAWs and electric fields (Faraon et al. 2010; Pagliano et al. 2014) or all-optical schemes (Jin et al. 2014; Peinke et al. 2021)
- Routing of on-chip generated single photons has been reported in waveguide devices at a frequency up to approx. 2-3 MHz range (Midolo et al. 2017) using electro-optic tuning or in NEMS actuated suspended devices (Papon et al. 2019), also in the 1-2 MHz range.

We are well aware that silicon based passive devices (i.e. without integrated emitters using off-chip sources) already reached a higher performance level in terms of losses, switching fidelity, reproducibility in fabrication etc, including SAW tuning (Kittlaus et al. 2021). However, this is not the type of system to compare to ours because it is not a monolithic platform with quantum emitters. As we now clearly point out in the paper, heterointegration strategies are on the way to bring high quality quantum emitters to passive platforms, including above mentioned silicon but also LiNbO₃ where first reports appeared for IPCs (Aghaeimeibodi et al. 2018) and SAW devices for instance by us.

These points of novelty, which were not fully clear in the previous version, are now clarified.

Changes made:

- In addition to the changes exemplified above, we also explicitly address the tuning range on page 13, line 389:

comparable to recent reports of static Stark tuning on this platform⁷⁸.

- We completely revised the first part of the introduction paragraph (page 2, lines 46-66) to improve the framing of our work and our method within the state of the art:

For most applications, the on-chip generation of single photons is highly desirable to avoid the inevitable coupling losses when using an off-chip source is used. For this purpose, quantum emitters¹⁴, for instance semiconductor quantum dots (QDs)^{15,16} or defect centers¹⁷⁻²², are excellent candidate systems. Among these systems, QDs made from (In,Ga,Al)As semiconductor compounds offer several advantages²³⁻²⁷: they are extremely bright sources of single photons^{28,29} and entangled photon pairs^{30,31} which can be elegantly included in the photonic structure during epitaxial growth of a heterostructure³². These heterostructures are then ready for monolithic fabrication of IQPCs using advanced cleanroom technology³³⁻³⁷. In contrast to monolithic approaches, heterogeneous integration of these QDs and other types of quantum emitters on IQPCs made on material platforms with complementary strengths promise superior performance¹². Such hybrid devices have been reported on silicon (Si)^{38,39},

silicon nitride (SiN)^{40–44}, aluminium nitride (AlN)⁴⁵ or lithium niobate (LiNbO₃)⁴⁶ IQPCs. However, their fabrication is natively connected to a significant increase of the complexity compared to monolithic routes. Furthermore, the on-chip control of light propagation in photonic elements is crucial. To this end, for instance thermo-optic^{47–49}, electro-optic^{50–52} or acousto-optic^{53–56} effects or nanomechanical actuation^{57,58} have proven to be viable routes. Among these mechanisms, acoustic phonons are an attractive choice because they couple to literally any system⁵⁹ enabling strong optomechanical modulation and dynamic reconfiguration of quantum emitters^{60–62}. In the form of radio frequency Rayleigh surface acoustic waves (SAWs)⁶³ or Lamb waves⁶⁴, phonons can be routed on-chip^{65–67} and interfaced with integrated photonic elements^{53–56,68,69}, quantum emitters^{70–74} or even superconducting quantum devices^{75,76}.

Comment 3.3:

Therefore, the authors have not convinced me on the potential scalability and/or novelty of their approach. At the current stage, I cannot recommend publication in Nature Communication.

Reply 3.3:

We hope that we made compelling points that our work is not just the presumed trivial combination of two concepts but indeed marks an important achievement deserving publication in Nature Communication.

Additional changes made:

- The data availability statement on page 15, lines 481/482 is amended. It will be updated during production to provide full details on where raw data used to produce the figures of the paper can be accessed.

References

- Aghaeimeibodi, Shahriar, Boris Desiatov, Je-Hyung Kim, Chang-Min Lee, Mustafa Atabey Buyukkaya, Aziz Karasahin, Christopher J. K. Richardson, Richard P. Leavitt, Marko Lončar, and Edo Waks, 2018, "Integration of Quantum Dots with Lithium Niobate Photonics," *Applied Physics Letters* **113**, 221102.
- Crespo-Poveda, A., A. Hernández-Mínguez, B. Gargallo, K. Biermann, A. Tahraoui, P. v. Santos, P. Muñoz, A. Cantarero, and M. M. de Lima, 2015, "Acoustically Driven Arrayed Waveguide Grating," *Optics Express* **23**, 21213.
- Davanco, Marcelo, Jin Liu, Luca Sapienza, Chen-Zhao Zhang, José Vinícius de Miranda Cardoso, Varun Verma, Richard Mirin, Sae Woo Nam, Liu Liu, and Kartik Srinivasan, 2017, "Heterogeneous Integration for On-Chip Quantum Photonic Circuits with Single Quantum Dot Devices," *Nature Communications* **8**, 889.
- Elshaari, Ali W., Efe Büyükközer, Iman Esmail Zadeh, Thomas Lettner, Peng Zhao, Eva Schöll, Samuel Gyger, et al., 2018, "Strain-Tunable Quantum Integrated Photonics," *Nano Letters (American Chemical Society)* **18**, 7969–7976.

- Elshaari, Ali W., Iman Esmaeil Zadeh, Andreas Fognini, Michael E. Reimer, Dan Dalacu, Philip J. Poole, Val Zwiller, and Klaus D. Jöns, 2017, "On-Chip Single Photon Filtering and Multiplexing in Hybrid Quantum Photonic Circuits," *Nature Communications* **8**, 379.
- Errando-Herranz, Carlos, Eva Schöll, Raphaël Picard, Micaela Laini, Samuel Gyger, Ali W. Elshaari, Art Branny, et al., 2021, "Resonance Fluorescence from Waveguide-Coupled, Strain-Localized, Two-Dimensional Quantum Emitters," *ACS Photonics* **8**, 1069–1076.
- Faraon, Andrei, Arka Majumdar, Hyochul Kim, Pierre Petroff, and Jelena Vučković, 2010, "Fast Electrical Control of a Quantum Dot Strongly Coupled to a Photonic-Crystal Cavity," *Physical Review Letters (American Physical Society)* **104**, 047402.
- Imany, Poolad, Zixuan Wang, Ryan A. DeCrescent, Robert C. Boutelle, Corey A. McDonald, Travis Autry, Samuel Berweger, et al., 2022, "Quantum Phase Modulation with Acoustic Cavities and Quantum Dots," *Optica* **9**, 501.
- Jin, Chao-Yuan, Robert John, Milo Y Swinkels, Thang B Hoang, Leonardo Midolo, Peter J van Veldhoven, and Andrea Fiore, 2014, "Ultrafast Non-Local Control of Spontaneous Emission.," *Nat Nanotechnol (Nature Publishing Group)* **9**, 886–90.
- Kittlaus, Eric A., William M. Jones, Peter T. Rakich, Nils T. Otterstrom, Richard E. Muller, and Mina Rais-Zadeh, 2021, "Electrically Driven Acousto-Optics and Broadband Non-Reciprocity in Silicon Photonics," *Nature Photonics* **15**, 43–52.
- Midolo, Leonardo, Sofie L. Hansen, Weili Zhang, Camille Papon, Rüdiger Schott, Arne Ludwig, Andreas D. Wieck, Peter Lodahl, and Søren Stobbe, 2017, "Electro-Optic Routing of Photons from a Single Quantum Dot in Photonic Integrated Circuits," *Optics Express* **25**, 33514.
- Pagliano, Francesco, YongJin Cho, Tian Xia, Frank van Otten, Robert John, and Andrea Fiore, 2014, "Dynamically Controlling the Emission of Single Excitons in Photonic Crystal Cavities.," *Nat Commun (Nature Publishing Group)* **5**, 5786.
- Papon, Camille, Xiaoyan Zhou, Henri Thyrrestrup, Zhe Liu, Søren Stobbe, Rüdiger Schott, Andreas D. Wieck, Arne Ludwig, Peter Lodahl, and Leonardo Midolo, 2019, "Nanomechanical Single-Photon Routing," *Optica* **6**, 524.
- Peinke, Emanuel, Tobias Sattler, Guilherme M. Torelly, Patricia L. Souza, Sylvain Perret, Joël Bleuse, Julien Claudon, Willem L. Vos, and Jean-Michel Gérard, 2021, "Tailoring the Properties of Quantum Dot-Micropillars by Ultrafast Optical Injection of Free Charge Carriers," *Light: Science & Applications* **10**, 215.
- Reithmaier, G., M. Kaniber, F. Flassig, S. Lichtmannecker, K. Müller, A. Andrejew, J. Vučković, R. Gross, and J. J. Finley, 2015, "On-Chip Generation, Routing, and Detection of Resonance Fluorescence," *Nano Letters* **15**, 5208–5213.
- Reithmaier, G., S. Lichtmannecker, T. Reichert, P. Hasch, K. Müller, M. Bichler, R. Gross, and J. J. Finley, 2013, "On-Chip Time Resolved Detection of Quantum Dot Emission Using Integrated Superconducting Single Photon Detectors," *Scientific Reports* **3**, 1901.
- Sarabalis, Christopher J., Timothy P. McKenna, Rishi N. Patel, Raphaël van Laer, and Amir H. Safavi-Naeini, 2020, "Acousto-Optic Modulation in Lithium Niobate on Sapphire," *APL Photonics* **5**, 086104.
- Schnauber, Peter, Jan Große, Arseny Kaganskiy, Maximilian Ott, Pavel Anikin, Ronny Schmidt, Sven Rodt, and Stephan Reitzenstein, 2021, "Spectral Control of Deterministically Fabricated Quantum Dot Waveguide Systems Using the Quantum Confined Stark Effect," *APL Photonics* **6**, 050801.
- Schnauber, Peter, Johannes Schall, Samir Bounouar, Theresa Höhne, Suk-In Park, Geun-Hwan Ryu, Tobias Heindel, et al., 2018, "Deterministic Integration of Quantum Dots into On-

- Chip Multimode Interference Beamsplitters Using in Situ Electron Beam Lithography," *Nano Letters* **18**, 2336–2342.
- Shao, Linbo, Mengjie Yu, Smarak Maity, Neil Sinclair, Lu Zheng, Cleaven Chia, Amirhassan Shams-Ansari, et al., 2019, "Microwave-to-Optical Conversion Using Lithium Niobate Thin-Film Acoustic Resonators," *Optica* **6**, 1498.
- Tadesse, Semere Ayalew, and Mo Li, 2014, "Sub-Optical Wavelength Acoustic Wave Modulation of Integrated Photonic Resonators at Microwave Frequencies," *Nature Communications* **5**, 5402.
- Tonndorf, Philipp, Osvaldo del Pozo-Zamudio, Nico Gruhler, Johannes Kern, Robert Schmidt, Alexander I. Dmitriev, Anatoly P. Bakhtinov, et al., 2017, "On-Chip Waveguide Coupling of a Layered Semiconductor Single-Photon Source," *Nano Letters* **17**, 5446–5451.
- Vogele, Anja, Maximilian M. Sonner, Benjamin Mayer, Xueyong Yuan, Matthias Weiß, Emeline D. S. Nysten, Saimon F. Covre da Silva, Armando Rastelli, and Hubert J. Krenner, 2020, "Quantum Dot Optomechanics in Suspended Nanophononic Strings," *Advanced Quantum Technologies* **3**, 1900102.
- Wan, Noel H., Tsung-Ju Lu, Kevin C. Chen, Michael P. Walsh, Matthew E. Trusheim, Lorenzo de Santis, Eric A. Bersin, et al., 2020, "Large-Scale Integration of Artificial Atoms in Hybrid Photonic Circuits," *Nature (Nature Research)* **583**, 226–231.
- Weiß, M., S. Kapfinger, T. Reichert, J. J. Finley, A. Wixforth, M. Kaniber, and H. J. Krenner, 2016, "Surface Acoustic Wave Regulated Single Photon Emission from a Coupled Quantum Dot–Nanocavity System," *Applied Physics Letters (AIP Publishing)* **109**, 033105.
- Weiß, Matthias, Daniel Wigger, Maximilian Nägele, Kai Müller, Jonathan J. Finley, Tilmann Kuhn, Paweł Machnikowski, and Hubert J. Krenner, 2021, "Optomechanical Wave Mixing by a Single Quantum Dot," *Optica* **8**, 291.
- Widhalm, Alex, Sebastian Krehs, Dustin Siebert, Nand Lal Sharma, Timo Langer, Björn Jonas, Dirk Reuter, Andreas Thiede, Jens Förstner, and Artur Zrenner, 2021, "Optoelectronic Sampling of Ultrafast Electric Transients with Single Quantum Dots," *Applied Physics Letters* **119**, 181109.
- Widhalm, Alex, Amlan Mukherjee, Sebastian Krehs, Nandlal Sharma, Peter Kölling, Andreas Thiede, Dirk Reuter, Jens Förstner, and Artur Zrenner, 2018, "Ultrafast Electric Phase Control of a Single Exciton Qubit," *Applied Physics Letters* **112**, 111105.
- Wigger, Daniel, Matthias Weiß, Michelle Lienhart, Kai Müller, Jonathan J. Finley, Tilmann Kuhn, Hubert J. Krenner, and Paweł Machnikowski, 2021, "Resonance-Fluorescence Spectral Dynamics of an Acoustically Modulated Quantum Dot," *Physical Review Research (American Physical Society)* **3**, 033197.
- Zadeh, Iman Esmaeil, Ali W. Elshaari, Klaus D. Jöns, Andreas Fognini, Dan Dalacu, Philip J. Poole, Michael E. Reimer, and Val Zwiller, 2016, "Deterministic Integration of Single Photon Sources in Silicon Based Photonic Circuits," *Nano Letters* **16**, 2289–2294.